# Drive the Dirac electrons into Cooper pairs in Sr$_x$Bi$_2$Se$_3$

Guan Du[1], Jifeng Shao[2], Xiong Yang[1], Zengyi Du[1], Delong Fang[1], Jinghui Wang[1], Kejing Ran[1], Jinsheng Wen[1,3], Changjin Zhang[2,3], Huan Yang[1,3], Yuheng Zhang[2,3] & Hai-Hu Wen[1,3]

Topological superconductors are a very interesting and frontier topic in condensed matter physics. Despite the tremendous efforts in exploring topological superconductivity, its presence is however still under heavy debate. The Dirac electrons have been proven to exist on the surface of a topological insulator. It remains unclear whether and how the Dirac electrons fall into Cooper pairing in an intrinsic superconductor with the topological surface states. Here we show the systematic study of scanning tunnelling microscope/spectroscopy on the possible topological superconductor Sr$_x$Bi$_2$Se$_3$. We first demonstrate that only the intercalated Sr atoms can induce superconductivity. Then we show the full superconducting gaps without any in-gap density of states as expected theoretically for a bulk topological superconductor. Finally, we find that the surface Dirac electrons will simultaneously condense into the superconducting state within the superconducting gap. This vividly demonstrates how the surface Dirac electrons are driven into Cooper pairs.

[1] National Laboratory of Solid State Microstructures and Department of Physics, Nanjing University, Nanjing 210093, China. [2] High Magnetic Field Laboratory, Chinese Academy of Sciences and University of Science and Technology of China, Hefei 230026, China. [3] Collaborative Innovation Center of Advanced Microstructures, Nanjing University, Nanjing 210093, China. Correspondence and requests for materials should be addressed to C.Z. (email: zhangcj@hmfl.ac.cn) or to H.Y. (email: huanyang@nju.edu.cn) or to H.-H.W. (email: hhwen@nju.edu.cn).

Topological insulators (TIs) are characterized by insulating bulk states and topologically protected metallic surface states because of the bulk band inversion[1,2]. The topological surface states have a linearly dispersing band structure which obeys the massless Dirac equation[1,2], and thus these electrons are called Dirac electrons. Because of finite spin-orbit coupling, the electrons of these surface states are spin nondegenerate and exhibit the spin-momentum locking effect.

After the demonstration of topological insulators, the search for topological superconductors (TSCs) has already become a hot topic in condensed matter physics[2], in one dimensional[3,4], two-dimensional (2D)[5,6] and three-dimensional (3D) systems[7–19]. Theoretical criterion for defining a 3D topological superconductor was proposed[11] and several materials were supposed or partly proved to be the candidates of 3D TSC, such as doping induced superconductors (SCs) from topological insulators[11,12] and topological crystalline insulators[15]. It is predicted that the surface of 3D TSC exhibits the topologically protected in-gap states which are known as the Majorana fermions. Point contact tunnelling measurements have detected zero-bias conductance peaks which are interpreted as the signatures of Majorana fermions[9,10,15] of TSC. However, the STM studies of $Cu_xBi_2Se_3$ (ref. 14) and $(Pb_{0.5}Sn_{0.5})_{0.7}In_{0.3}Te$ (ref. 19) show full gaps and give opposite views about the presence of topological superconductivity in these systems. Alternatively, it was proposed that 2D topological superconductivity may be induced by the superconducting proximity effect and a Majorana fermion will emerge in the vortex core[20]. This interesting proposal seems to be getting support from the experiments of heterostructures with a conventional SC and a TI (refs 5,6). Because of the peculiar spin-momentum locking effect mentioned above, the superfluid with the spin singlet Cooper paring is not completely comforted by the Dirac electrons in superconducting state. Therefore, it is highly desired to know whether and how these Dirac electrons are driven into Cooper pairs. As far as we know, a vivid demonstration of driving the surface Dirac electrons into Cooper pairs, especially in an intrinsic system with both bulk superconductivity and topological surface states, is still lacking.

Recently, superconductivity has been discovered in $Sr_xBi_2Se_3$ which is supposed to be a promising candidate of TSC (refs 21–24). Experimentally the quantum oscillation with the Dirac dispersion has been detected in the measurement of global resistivity in the normal state[21], which is supported by the data of angle resolved photo-emission spectroscopy (ARPES)[23]. The existence of topological surface states, lower Fermi level, more robust bulk superconductivity compared to $Cu_xBi_2Se_3$ make $Sr_xBi_2Se_3$ a good platform to study the possible topological superconductivity[21].

Here we present scanning tunnelling microscope/spectroscopy (STM/STS) studies on the newly discovered superconductor $Sr_xBi_2Se_3$. Our results give three important insights toward those issues quoted above. Firstly, the topological surface state and superconducting spectrum with a deep suppression of the density of states (DOS) within the gap are observed for $Sr_xBi_2Se_3$. Secondly, it is on an intrinsic system, that is, $Sr_xBi_2Se_3$, not by the TI/SC heterostructures, that we observe the superconductivity and the topological surface states. The spectrum with high quality observed by our experiment indicates the absence of the in-gap states. Thirdly, the most important point is the clear demonstration of driving the Dirac electrons into Cooper pairs. This is vividly shown by the clear weakening of the quantum oscillations due to Landau levels (LLs) within the superconducting gap. We present the straightforward evidence that the surface Dirac electrons will simultaneously condense into the superconducting state when the energy is smaller than the bulk superconducting

gap. This observation will stimulate the exploration of the unique properties of the topological superconductor and the Majorana fermions in an intrinsic system.

## Results

**Topography and spectroscopy studies of $Sr_xBi_2Se_3$.** We have grown the superconducting $Sr_xBi_2Se_3$ crystals using flux method[21] and conducted systematic studies using STM/STS. The nominal doping level $x$ is about 0.2. The samples generally exhibit sharp superconducting transitions which are comparable with previous reports[21,22,24], and the characterization of the basic properties is presented in Supplementary Fig. 1. Our samples, like $Cu_xBi_2Se_3$, display clear inhomogeneity in general. In our STM/STS studies, two kinds of regions with significant distinctions are found. The first kind has a very clean and atomically flat surface, as shown in Fig. 1a. The impurities can be viewed as substituted Sr atoms and show up as triangular like images[23], as viewed by the atomically resolved topography in Fig. 1a,c. As presented in Fig. 1b, the tunnelling spectrum obtained in this region does not possess an evident superconducting gap at low-temperatures. The local height of the impurity is only slightly enhanced (Fig. 1a) and the tunnelling spectrum without superconducting gapped feature is quite general in this region, and varies little when traverse crossing the substituted Sr, as shown by the bottom panel of Fig. 1c. The second kind of region has terraces decorated by clusters with much larger height scattering when crossing them, as shown in the middle of Fig. 1d. Those clusters vary in size and height and mostly are movable by the STM tip. We presume that the clusters are composed by intercalated or inserted Sr atoms, just like the Cu clusters in $Cu_xBi_2Se_3$ (refs 7,14). Concerning the difficulty of intercalating Sr atoms with much larger radii into the van der Waals spacing, it may be possible that Sr clusters are formed in the growing process with fault growth along the $c$-axis, and the cleaving is easier to occur at this layer. Interestingly, in a region with many Sr clusters, the tunnelling spectrum exhibits a clear superconducting gapped feature (Fig. 1e). Substituted Sr atoms also exist in the superconducting region and turn out to have little effect on superconducting gap (Fig. 1f). We assume that the Sr substitution, probably doped to the Bi sites, behaves as a hole doping concerning the different cationic states of $Sr^{2+}/Bi^{3+}$, while the Sr clusters remaining in the van der Waals gap act as electron doping. If this scenario is correct, it means that electron doping by the intercalated Sr can induce superconductivity in $Bi_2Se_3$, as in the case of $Cu_xBi_2Se_3$.

**Studies on the superconducting gap.** According to the theoretical proposal for TSCs[11], $Sr_xBi_2Se_3$ is considered to be a promising candidate. To verify its topological property, analysing the superconducting pairing symmetry is essential. In the region with well-formed superconductivity, we manage in detecting the tunnelling spectrum with the bottom almost touching zero near zero-bias energy. The Sr clusters in big size can be viewed everywhere in the region shown in Fig. 2a. The blurry transverse lines in Fig. 2a are caused by dragging the clusters during scanning, demonstrating the mobility of the Sr clusters. When going across in this region, as highlighted by the white arrowed line in Fig. 2a, the shape of the tunnelling spectra and the gap values are quite homogeneous according to the spectra measured at 400 mK, as shown in Fig. 2b. The spectrum displayed in Fig. 2c is the average of all the data in Fig. 2b. The tunnelling spectra with pronounced coherence peaks and perfect suppression of $dI/dV$ within the gap look much better than the previous STM/STS results in $Sr_xBi_2Se_3$ (ref. 23). The U shape instead of a V shape feature suggests that the superconducting gap is nodeless. The DOS within the superconducting gap is fully gapped and the

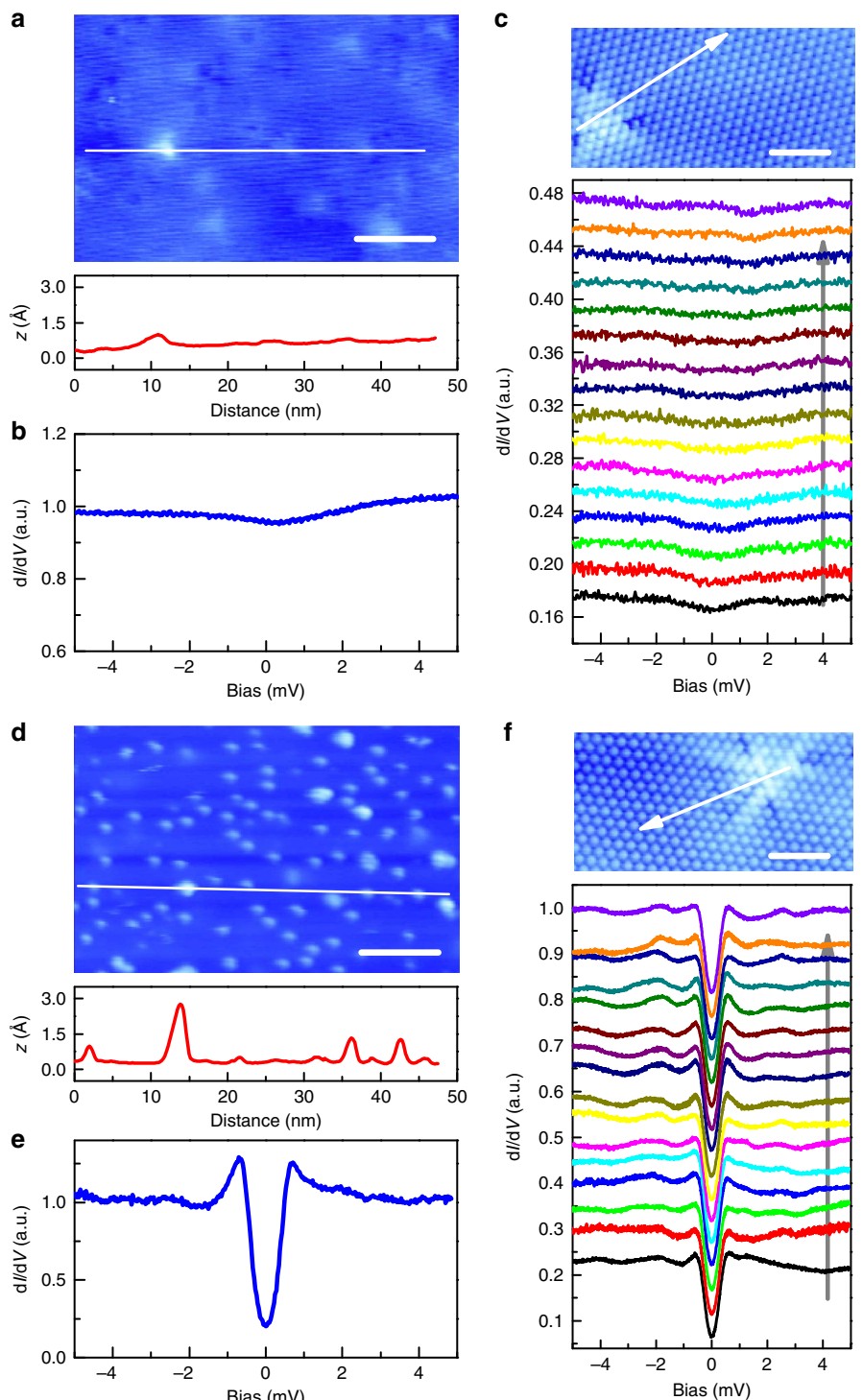

**Figure 1 | Topographic STM images and tunnelling spectra.** (**a,d**) Topographic image of the non-superconducting (**a**) or the superconducting (**d**) region with different density of Sr clusters. Scale bars, 10 nm. The center of a triangular image in **a** is roughly corresponding to a substituted Sr impurity. Substituted Sr also exists in the region with well-formed superconductivity with more Sr clusters. The curves in **a,d** show the height distribution measured along the white lines marked in corresponding topographies. (**b,e**) A typical tunnelling spectrum measured at 400 mK in non-superconducting (**b**) or superconducting (**e**) area with the corresponding topography shown in **a** or **d**, respectively. The tunnelling current ($I_t$) is 103 pA. (**c,f**) shows the atomic resolved topography of non-superconducting (**c**) and superconducting (**f**) region. Scale bars, 2 nm. The spatially resolved tunnelling spectra taken at zero field and along the arrowed lines are shown below the images measured with $I_t = 103$ pA.

remaining finite zero-bias DOS is because of a slight scattering/broadening effect[25]. There is no abnormal DOS near zero-bias energy, which contradicts the theoretical proposals of a 3D TSC (ref. 12).

To have a comprehensive understanding on the data, we fit the tunnelling spectrum with different gap functions. As shown in Fig. 2c,d, the spectrum can be nicely fitted by the Dynes model[25] using a combination of two components corresponding to two

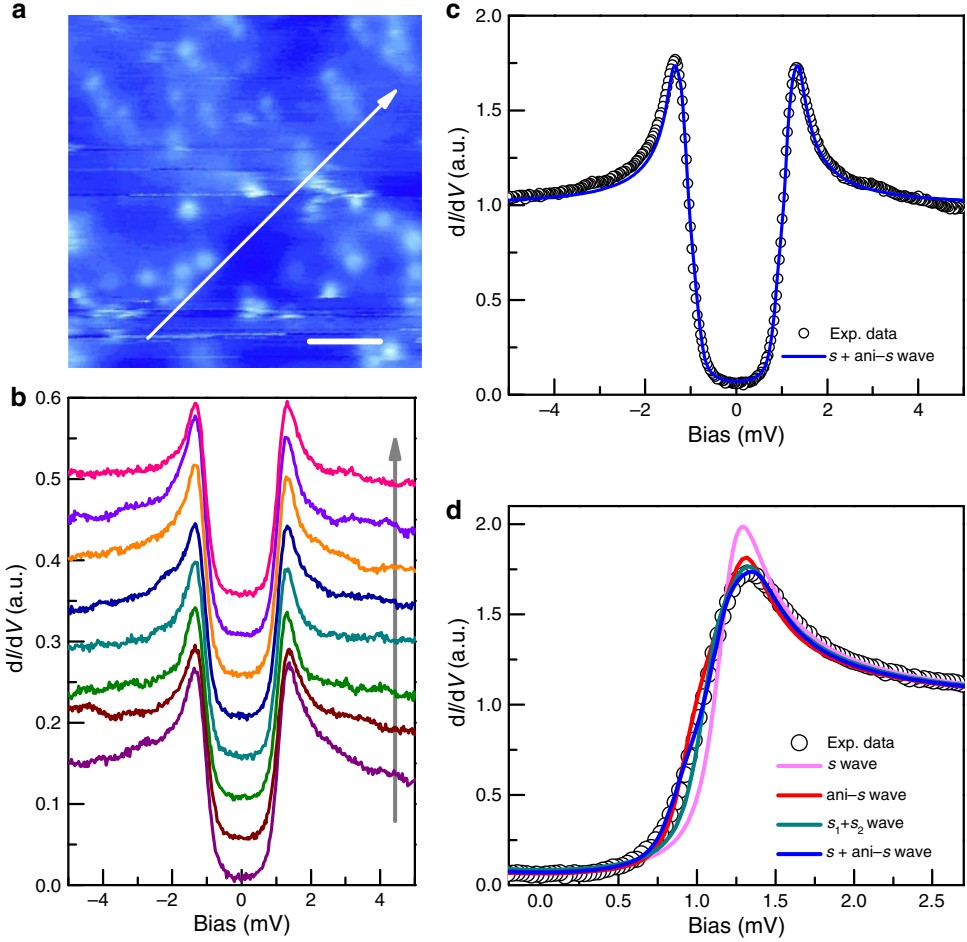

**Figure 2 | Full superconducting gaps and theoretical fittings.** (**a**) The topography of the superconducting area with quite dense Sr clusters. Scale bar, 10 nm. (**b**) The spatially resolved tunnelling spectra taken from a series of points equally spaced along the arrowed line in **a** at 400 mK, $I_t = 102$ pA. (**c**) The averaged and normalized spectrum of the spectra in **b**. The solid curve is the theoretical fitting with Dynes model using two components (*s* wave and ani-*s* wave). (**d**) The comparison of theoretical fittings with different superconducting gap functions.

gaps, one with an *s* wave and another with a slightly anisotropic *s* wave gap, with $\Delta_s = 1.15$ meV and $\Delta_{\text{ani}-s} = 1.37(0.19 \cos 4\theta + 0.81)$ (meV). As a comparison, we fit the data with several other scenarios of different superconducting gaps, as shown in Fig. 2d. All the parameters used for fitting are obtained by minimizing the deviation between the fitting curve and the raw data. The fitting curves shown in Fig. 2d are the optimized ones for each model. We further calculate the difference values between the raw spectrum and the fitting curves of four fitting models, and the results are shown in Supplementary Fig. 3 and described in Supplementary Note 1. We thus conclude that the fitting with two components associated with double gaps (*s* + ani-*s* wave) can interpret the nature of superconductivity in the material very well. We have also conducted studies by adjusting the fraction of the two components, and a combination of the *s* wave with the weight of 26% and the ani-*s* wave gap with the weight of 74% turns out to give the best fit (Fig. 2c,d). The detailed fitting results are given in Supplementary Note 1 (see also Supplementary Figs 2–5 and Supplementary Table 1). These results suggest that there should be two bands with different gaps existing at the Fermi energy, which is consistent with the recent ARPES study of $Sr_xBi_2Se_3$ that the topological surface state coexists with the bulk state around the Fermi energy[23]. According to the ARPES data, the Fermi surfaces of $Sr_xBi_2Se_3$ are two centric circles with similar size around $\bar{\Gamma}$ point (with probably a slight

hexagonal distortion)[8]. Because of the spin non-degeneracy, the DOS of the topological surface state on Fermi surface is about half of that in bulk state. Taking the *z*-axis dispersion of the bulk state into account, the proportion of the DOS of the two components on the Fermi surface between topological surface electrons and normal bulk electrons should be <33% and >67%, respectively. From above analysis, we can conclude that the ani-*s* wave gap with the weight ratio of 74% may be contributed by bulk superconductivity, and the *s* wave gap with the ratio of 26% is because of the Dirac electrons of the topological surface state in the superconducting state. Full gapping of tunnelling spectrum manifests itself that the Dirac electrons of topological surface state have been driven into Cooper pairs.

**Temperature and field dependence of superconductivity.** As displayed in Fig. 3a, we further measure tunnelling spectra at different temperatures. The gap values are different from that in Fig. 2 since the measurements are made at different positions with different local conditions. The superconducting feature vanishes at about $T_c \approx 5$ K, which is higher than that in the transport and magnetization measurements (Supplementary Fig. 1a). The tunnelling spectra at different temperatures can be well fitted by the two-component (gap) model (*s* wave and ani-*s* wave gaps) with the fixed proportion as quoted above. As shown

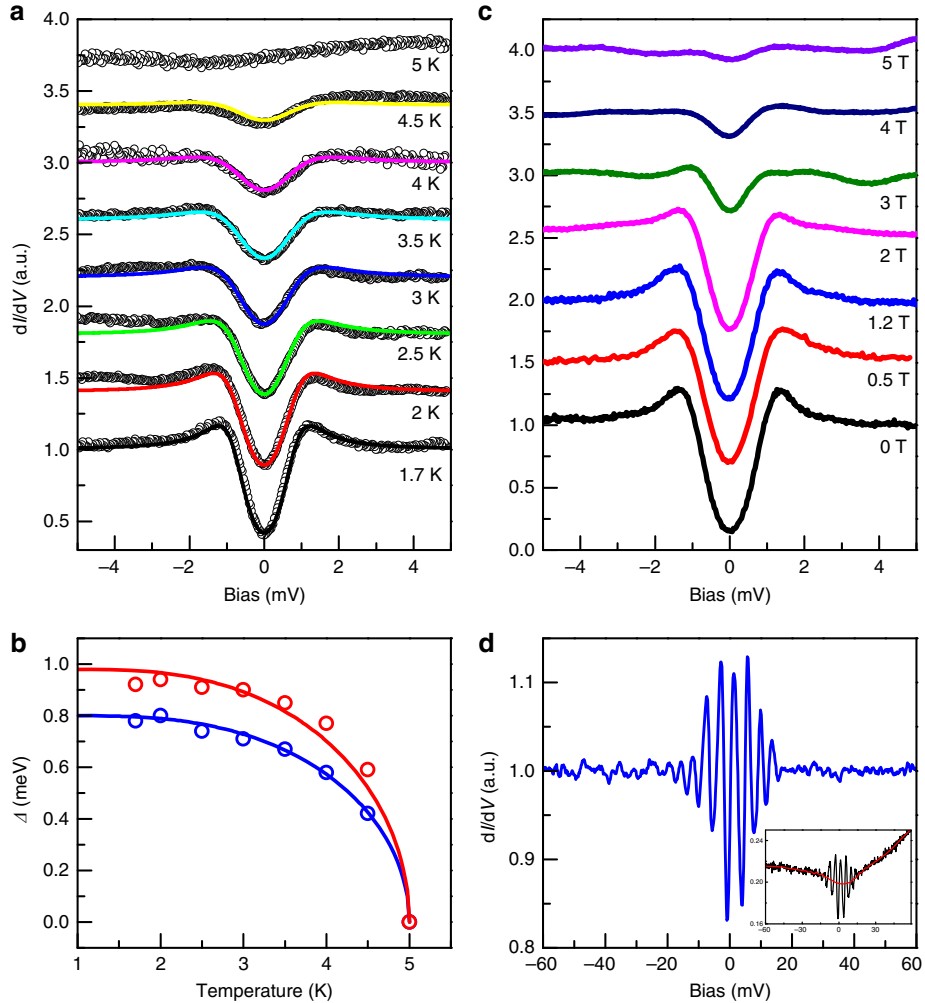

**Figure 3 | Temperature and magnetic field dependence of tunnelling spectra and Landau levels.** (**a**) The evolution of the tunnelling spectra with temperatures increased from 1.7 to 5 K at zero field, $I_t = 103$ pA. The superconducting gapped feature vanishes gradually at about 5 K. The solid lines are theoretical fittings using two components (s wave and ani-s wave). (**b**) The temperature dependence of $\Delta_s$ (blue dots) and $\Delta_{\text{ani}-s}$ (red dots) obtained by fitting. The solid lines are obtained through the numerical solution to the BCS gap equation by fixing $\Delta_s(0) = 0.8$ meV, $\Delta_{\text{ani}-s}(0) = 0.98$ meV and $T_c = 5$ K derived from the fitting results in **a**. (**c**) Evolution of the tunnelling spectra with magnetic field increased from 0 to 5 T, $I_t = 103$ pA. Here for each field, the tunnelling spectrum is obtained by averaging the STS data measured at different locations, therefore the oscillations because of the LLs is not visible. (**d**) The d$I$/d$V$ spectrum measured at 5 T at a fixed location (smoothed by averaging about 20 STS curves measured at the same location) with the background subtracted. The spectrum has been smoothed by averaging the neighbouring 10 data points to lower down the noise. The black curve of inset shows the raw data. The background is obtained by averaging the neighbouring 200 data points shown as the red curve in the inset, $I_t = 102$ pA.

in Fig. 3b, the temperature dependent gap values derived from the fitting to the Dynes model obey the BCS theoretical curve. Figure 3c shows the magnetic field dependence of tunnelling spectra verifying that the $\mu_0 H_{c2}$ can be as high as 5 T which is also larger than that in the transport measurements (Supplementary Fig. 1d). Note the tunnelling spectra shown here at different magnetic fields are obtained by averaging many tunnelling spectra measured at different locations in the same area under the same field, therefore the oscillations of d$I$/d$V$ because of the LL effect are not seen here.

**Observation of Landau levels**. With a magnetic field of 5 T, as shown in Fig. 3d, we succeed in observing the LL peaks near the Fermi energy in a clean region over the length scale of about 300 nm with many surrounding Sr clusters. This tunnelling spectrum is obtained at a fixed location. As has been systematically studied in plenty of works, the LLs of 2D surface state can

be easily viewed by tunnelling spectra while the 3D bulk state LLs are absent[26–30]. It is thus reasonable to conclude that the LLs observed here originate from the topological surface state[26–30]. The enhancement of the LL peaks near Fermi energy is because of the much enhanced quasiparticle lifetime of the surface states in approaching the Fermi energy[26]. We have further measured the tunnelling spectra at magnetic fields of 0, 2, 3, 4 and 5 T in the same area and the normalized data are shown in Fig. 4a–e. At each magnetic field, the spectra are taken from a series of points equally spaced along a line with the length of about 200 nm. The LL oscillations are obvious in the spectra taken with magnetic field especially with the field of 5 T. We obtained the bias values of the LL peaks from the spectrum in Fig. 3d and some of the spectra with obvious LL oscillations in Fig. 4e, and fitted the $n$th LL peak energies ($E_n$) by using the equation of $E_n = E_D + v_F\sqrt{2eB\hbar|n|}$ with $E_D$ the Dirac point energy and $v_F$ the Fermi velocity, as shown in Supplementary Note 2. Although there is an uncertainty in the fitting results shown in

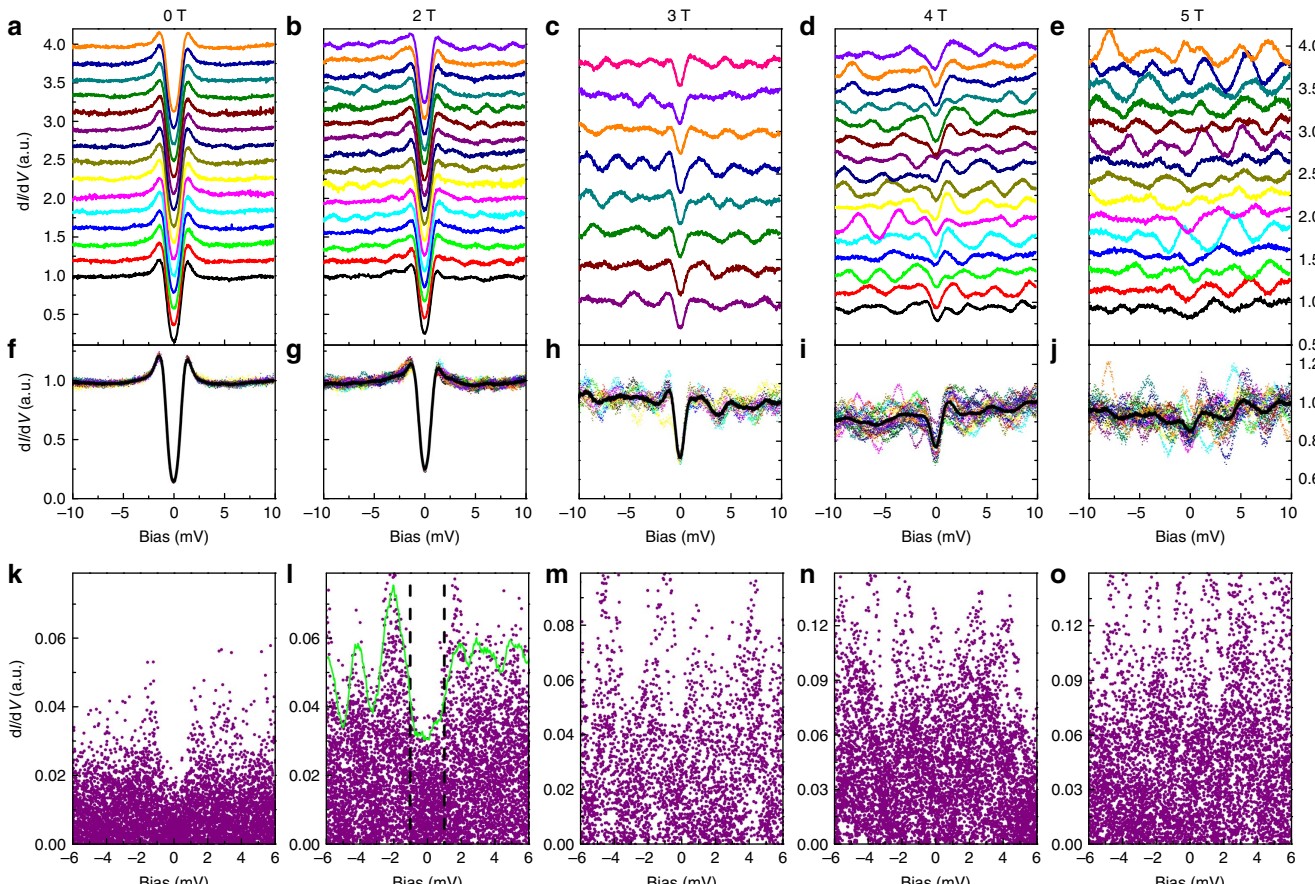

**Figure 4 | The variation of LL oscillations with magnetic field. (a–e)** The spatially resolved spectra obtained over the same area with the magnetic field of 0, 2, 3, 4 and 5 T, respectively, $I_t = 103$ pA. **(f–j)** The experimental data plotted together with the average spectrum at each magnetic field. The black curves are the averaged spectra. **(k–o)** The absolute value of the differences between the normalized spectrum at different locations and the averaged curve, this exemplifies only the fluctuation because of the LL oscillations. The green line in **l** is the envelop curve which is defined as 2.5 times of the mean value of the LL oscillations. The mean value is calculated by averaging the data in the region of ( − 0.2 mV, 0.2 mV) around each bias voltage. The dashed vertical lines in **l** show the approximate gap value at 2 T. One can see that the oscillations because of the LLs are strongly suppressed within the gap, which can be seen from both the raw data in **b,g** and the LL oscillations in **l**.

Supplementary Fig. 6, which is resulted from the deviations of the obtained LL bias values as discussed in Supplementary Note 2, we can still get the approximate values of $E_D$ and $\nu_F$ by averaging the fitting parameters of different sets of data to lower down the uncertainty. The determined value of $E_D$ is about − 340 meV which is consistent with the ARPES study of $Sr_xBi_2Se_3$ (ref. 23), and $\nu_F$ is about $6.7 \times 10^5$ m s$^{-1}$, which is comparable with the STS and ARPES study of $Bi_2Se_3$, that is, $\nu_F = 3.4 \times 10^5$ m s$^{-1}$ by STS study[27] and $\nu_F = 5 \times 10^5$ m s$^{-1}$ by ARPES study[31].

We also notice that there are some fine structures with much smaller amplitude accompanying with the pronounced LL peaks, which can be visualized in Fig. 4e. Similar secondary fine structures were also observed in the Landau level studies of $Bi_2Se_3$ by STM (ref. 26) and transport measurements[32], and the authors there concluded that the pronounced LL peaks are originated from the topologically protected surface state. The fine structures in our present work may have the same origin as theirs. The ARPES study[33] on $Bi_2Se_3$ shows that the bulk states can also form quasi-2D electron gas near the surface because of the bending of the bulk conduction band in approaching the surface. We can now probably attribute these fine structures accompanied with the LL peaks on tunnelling spectra to the bulk-derived 2D electron states. But clearly they are much weaker comparing with the intensity of the main structure arising from the LL peaks of the surface Dirac electrons. To summarize, we can naturally

conclude that the LL oscillations observed here originate from the topologically protected surface state of $Sr_xBi_2Se_3$ and the surface Dirac electrons stay intact and do not merge into the bulk band at Fermi level. This along with the two-component model fitting referred above makes it plausible that the surface Dirac electrons have been driven into superconductivity, which may be caused by the proximity effect of the bulk superconductivity.

**Driving the Dirac electrons into Cooper pairs.** Since the Fermi energy is within the conduction band of the bulk state, the Dirac point is quite far away from the Fermi level (about 340 meV, see Supplementary Note 2). Therefore, any tiny spatially local alteration to the electronic state, like the in-plane stress which might mildly influence the dispersion of the surface state, will shift the LL peak positions[29]. The surface of this area has a slight fluctuation which can be caused by the intercalated Sr beneath the surface layer leading to in-plane stress, or the inhomogeneous doping level of electrons. Therefore, the random behaviour of LL peaks over a large region can be viewed and understandable. This can offer us the opportunity to detect the information of Dirac electrons in the superconducting state by observing the amplitude of the LL fluctuations if the spectra measured at a fixed magnetic field are plotted together. To vividly show whether and how the Dirac electrons are driven into Cooper pairs in $Sr_xBi_2Se_3$, we did a

systematic research on the LL peaks in the spectra in Fig. 4. The data treating and normalizing process are presented in Supplementary Note 3. The enhancement of LL fluctuations and the suppression of superconductivity versus magnetic field can be seen clearly here. One can find that the tunnelling spectra are rather homogeneous and smooth at 0 T, as presented in Fig. 4a. However, at 2 T, a waving like signal of d$I$/d$V$ outside the superconducting gap region shows up, and it gets stronger with a higher magnetic field (Fig. 4b–e). These waving like signals are actually because of the LLs with a spatial fluctuation effect[29]. To see it more clearly, we take an average of all tunnelling spectra measured at the same magnetic field but different positions, and then overlay the data together with their averaged curve as shown by the black solid lines in Fig. 4f–j. One can clearly see that the spectra outside the superconducting gap oscillate around the average value and the oscillation magnitude increases with the magnetic field, indicating that the fluctuation is caused by the LLs rather than the intrinsic noise or background. We then subtract the average spectrum from the data measured at different locations, leaving the oscillations only and plot the absolute values together for each field, as shown in Fig. 4k–o. At zero field, as shown in Fig. 4k, the amplitude of oscillations is quite small. Here the slight difference of the oscillation amplitude between inside and outside the superconducting gap region is caused by the finite floating background of the spectra obtained at different positions. When the magnetic field is 2 T the oscillation amplitude outside the superconducting gap is getting much stronger, but that within the gap shrinks rapidly to a smaller value (see for example, Fig. 4l). Since the data at 2 T clearly exhibit both the superconducting gap and the LL oscillations, we calculated the mean value (MV) of LL oscillations in a region of (−0.2 mV,

0.2 mV) around each bias value, and we define the 2.5 times MV as the envelop curve as indicated by the green curve in Fig. 4l. One can find that the envelop curve within the gap in Fig. 4l is much lower than that at higher bias outside the gap region, indicating that the DOS of the Dirac electrons inside the superconducting gap are gapped away and they are driven into Cooper pairs. At a magnetic field of 3 T, the superconducting gap is still visible (Fig. 4h) and the oscillation amplitude is getting stronger compared with 2 T. Under the magnetic fields of 4 and 5 T, the gapped feature is hardly visible in the raw and overlaid data, which means that most of the Cooper pairs are broken. It is reasonable to take the amplitude of oscillations from Fig. 4l–o as the index of the population of the Dirac electrons. If the Dirac electrons stay unpaired, the amplitude of oscillations because of the LLs would be the same within and outside the gap. The shrinkage of the amplitude of LL oscillations within the gap at 2 T (probably also seen partly at 3 T) is consistent with the picture of driving the Dirac electrons of the surface topological state into Cooper pairs which simultaneously condense into the superconducting state.

We further calculated the mean values of the LL oscillations in Fig. 4k–o at each magnetic field. The mean values at different magnetic fields are determined through averaging the data in Fig. 4k–o in the window of (−1 mV, 1 mV) and (−10 mV, 10 mV), and are shown in Fig. 5a by open circles and open squares, respectively. It is clear that the mean value increases monotonously with the field, which is consistent with the properties of the LL oscillations. To present the magnetic field induced evolution of the density of Dirac electrons near Fermi level, we calculated the ratio of the mean values of the LL oscillations in the range of (−1 mV, 1 mV) and (−10 mV,

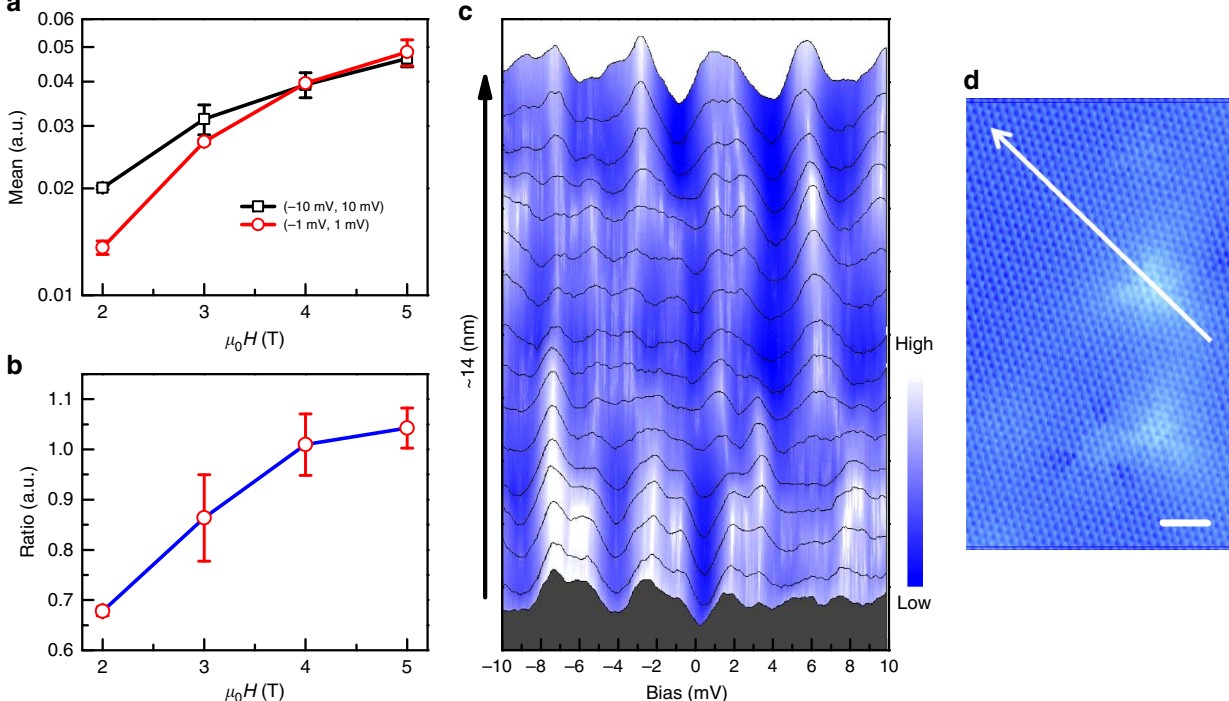

**Figure 5 | The magnetic field induced evolution and the spatial variation of the LL oscillations.** (**a**) The mean values of the LL oscillations calculated in the window from −10 to 10 mV (open squares) and −1 to 1 mV (open circles) at each magnetic field. (**b**) The ratio of the mean values of the LL oscillations averaged in the range of (−1 mV, 1 mV) and the range of (−10 mV, 10 mV). The error bars in **a,b** are defined as the differences between the mean values and those from the fitting to the Gaussian distribution, see Supplementary Note 5. (**c**) The spatially resolved tunnelling spectra taken from a series of points equally spaced along the arrowed line in **d** at 400 mK and 5 T in a different region, $I_t = 101$ pA. (**d**) The topography of the measured area for data in **c**. Scale bar, 2 nm.

10 mV) for each field, which is displayed in Fig. 5b. The error bars in Fig. 5a,b are determined as the differences between the mean values and those from the fitting to Gaussian distributions, see Supplementary Note 5. The ratio is small at 2 and 3 T indicating low density of Dirac electrons at Fermi level, and $\cong 1$ at 4 and 5 T. The clear suppression of LL oscillations within the gap at low field is most likely caused by the pairing and condensation of Dirac electrons. Along with the full superconducting gap at 0 T shown in Fig. 2c, our study gives the clear evidence of driving the Dirac electrons of the topological surface state into Cooper pairs.

**The spatial variation of the LLs**. Fig. 5c shows a series of spectra taken along a line across one substituted Sr impurity with the length of 14 nm. The points where the spectra are taken are equally spaced from the starting point to the end point of the white arrowed line indicated in Fig. 5d. Distinct LL peaks can be viewed in the spectra in the top of Fig. 5c accompanied by some fine structures as described above. Those spectra share the same feature in general. In the middle of Fig. 5c, the peaks of the spectra become less sharp. For the spectra taken near the impurity shown in the bottom of Fig. 5c, the peaks are out-of-shape and even disappear at the bias of 6 mV. We conclude that the LLs can be disturbed by the substituted Sr impurity. Such disturbance may be caused by the local in-plane stress, fluctuation of the local DOS, or the scattering effect near the Sr impurity.

## Discussion

For a 3D TSC, it is predicted that the surface Andreev bound state appears within the bulk quasiparticle gap giving rise to significant zero-bias conductance peak on the surface and making the coherence peaks suppressed or absent[12,13]. This contrasts sharply with our STM/STS results, indicating that $Sr_xBi_2Se_3$ may not be explained by that theory for a bulk TSC. From our theoretical fitting to the spectra in Fig. 2c, it can be concluded that the bulk component of the superconducting gap is slightly anisotropic. Recently, Matano et al.[34] observed a two-fold angle dependence of the Knight shift in the NMR measurements in $Cu_xBi_2Se_3$, which gets a qualitative support from the specific-heat measurements by Yonezawa et al.[35] This may be explained as the possible nematic superconducting state[36] or the triplet pairing in this material. Actually, in the two-component model with s and ani-s gaps, we used two different gap equations for the anisotropic s-wave gap, that is, $0.19 \cos 4\theta + 0.81$ and $0.35 \cos 2\theta + 0.65$, and the first one is a little better. However, if merely judging from the quality of the fitting, we could not rule out the two-fold gap functions. As shown in Supplementary Fig. 8 and addressed in Supplementary Note 4, the fitting result with an anisotropic gap is not inconsistent with the expectation of a two-fold component of the gap function.

Our experiment shows that the topological surface state persists independently from the bulk band at the Fermi level and the Dirac electrons are driven into Cooper pairs at low temperatures. We must emphasize that the induced Cooper pairs constructed by the Dirac electrons of the surface state may behave differently from the Cooper pairs in the bulk, since the former have a spin-momentum locking effect. In this case, a partial spin triplet component may be expectable[20]. This interesting expectation calls for future efforts. We can reasonably assume that the condensed Dirac electrons of the surface state discovered in this work may form a 2D TSC, which will spontaneously realize a Majorana mode in the vortex core that exhibit non-Abelian quantum statistics[20]. Our experiment performed in $Sr_xBi_2Se_3$ certainly makes a step further in helping to search for the TSC and the Majorana fermions in an intrinsic superconducting system.

## Methods

**STM/STS measurements**. For STM/STS measurements we cleaved the samples in an ultrahigh vacuum chamber with a base pressure of about $10^{-10}$ Torr. The STM studies are carried out with an ultrahigh vacuum, low-temperature and high-magnetic field scanning probe microscope USM-1300 (Unisoku Co., Ltd.) with pressure better than $10^{-10}$ Torr. The tunnelling spectra are measured by a lock-in amplifier with an AC modulation of 0.1 mV at 987.5 Hz in order to lower down the noise.

**Date availability**. The data that support the findings of this study are available from the corresponding authors on request.

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

## Acknowledgements

We thank Yong Chen at Purdue University, Qianghua Wang at Nanjing University, Genda Gu at Brookhaven National Lab for helpful discussions. This work was supported by the Ministry of Science and Technology of China (grant number: 2016YFA0300400), the National Natural Science Foundation of China (NSFC) with the projects: A0402/11534005, A0402/11190023, A0402/11374143, A0805/U1532267.

## Author contributions

The low-temperature STS measurements were performed by G.D., X.Y., Z.Y.D., D.L.F., H.Y. and H.-H.W. The samples were prepared by J.F., C.J.Z., J.H.W., J.S.W., K.J.R. and Y.H.Z. The simulation based on the Dynes model and the data analysis was performed by G.D. and X.Y.. G.D. and H.-H.W. wrote the manuscript which was supplemented by H.Y. H.-H.W. coordinated the whole work. All authors have discussed the results and the interpretations.

## Additional information

**Competing financial interests:** The authors declare no competing financial interests.

**Publisher's note**: 

