## [Peer Review File · Nature Communications]

Reviewers' comments:

Reviewer #1 (Remarks to the Author):

The authors have answered well most questions asked by both referees and I recommend its acceptance in Nature Comm. This work is a interesting and informative study on topological superconductivity, revealing the physics of surface electrons condensing into Cooper pairs.

Here are two remaining suggestions that I recommend the authors to take into account when preparing their finalized manuscript for publication:

1) I think both referees raised questions how the authors can be sure that the Landau levels arise from topological surface state Dirac fermions. The authors argued that STM measured LLs are unlikely to arise from 3D bulk carriers with z-dispersion, which is reasonable. However, the bulk states can also form quasi 2D states near the surface due to the bending of the bulk conduction band near the surface, giving rise to an inversion layer of 2D electrons. Existence of such bulk-derived 2D electron states has been documented in many papers on TIs, for example doi:10.1038/ncomms1131. I suggest the authors add some discussion to either rule out this possibility, or if they cannot, weaken their claim for the origin of LLs from absolute "smoking gun" to be "consistent with" TSS.

2) A possible typo in line 153: is it really Fig. 2c?

Reviewer #2 (Remarks to the Author):

The authors have substantially edited their manuscript and taken on board many of the suggestions. The manuscript has improved significantly, and there are only a few points which I think should still be accounted for. I recommend publication in Nature Communications after the suggested changes have been considered.

The discussion of percentage of density of states of gaps for isotropic s-wave and anisotropic s-wave is still confusing, I believe this is mainly due to the notation the authors use. I recommend introducing different symbols for the gap functions themselves (which are currently called Δ_s and $\Delta_{s\text{-ani}}$) and the associated densities of states.

A few points need clarification:

* The Landau levels themselves are not the smoking gun that the surface state becomes gapped, but much rather the fact that the gap goes to zero rather than to a finite value

* For the averaging of the spectra in fig. 5, two different energy ranges are mentioned: the main text says $[-1,1]$, whereas the caption indicates $[-0.5,0.5]$. This average should also be plotted in fig. 5a, together with error bars.

* In the caption of figure 4, "MV" should be defined.

* The additional figures in the supplementary material showing the various fits are very good. Fig. S4b-d make one wonder whether the fit with a slightly larger value of Gamma is maybe better than the one shown in fig. 2d ?

REVIEWERS' COMMENTS:

Reviewer #1 (Remarks to the Author):

The authors have answered all questions satisfactorily and I recommend publication of this paper in Nature Comm.

Reviewer #2 (Remarks to the Author):

The revised version of the manuscript has accounted for almost all changes both referees had suggested. One point which the authors probably overlooked but which I believe is important is that in fig. 5a a plot of the mean value of the averaged Landau level oscillations in the window $[-1\text{mV}, 1\text{mV}]$ (with error bars) is still missing (next to the one for the window $[-10\text{mV}, 10\text{mV}]$). To be able to follow the authors on how they obtain the ratio in fig. 5b this would be important. Given that this ratio is close to one, the mean value of these oscillations should fit rather nicely in fig. 5a.

With this point fixed, I recommend publication in Nature Communications.

Response to Reviewers' comments:

Words with blue color are the comments of the Reviewer. The responses are given in words in black color.

=====

Reviewer #1 (Remarks to the Author):

The authors have answered well most questions asked by both referees and I recommend its acceptance in Nature Comm. This work is a interesting and informative study on topological superconductivity, revealing the physics of surface electrons condensing into Cooper pairs.

Here are two remaining suggestions that I recommend the authors to take into account when preparing their finalized manuscript for publication:

1) I think both referees raised questions how the authors can be sure that the Landau levels arise from topological surface state Dirac fermions. The authors argued that STM measured LLs are unlikely to arise from 3D bulk carriers with z-dispersion, which is reasonable. However, the bulk states can also form quasi 2D states near the surface due to the bending of the bulk conduction band near the surface, giving rise to an inversion layer of 2D electrons. Existence of such bulk-derived 2D electron states has been documented in many papers on TIs, for example doi:10.1038/ncomms1131. I suggest the authors add some discussion to either rule out this possibility, or if they cannot, weaken their claim for the origin of LLs from absolute "smoking gun" to be "consistent with" TSS.

Response: We thank the referee for mentioning this point to us. Indeed, as illustrated by the ARPES study (doi:10.1038/ncomms1131), there might exist bulk-derived 2D electron states which coexist with the topological surface state on the surface. As described in the text and illustrated in related figures, in addition to the main structure of LL peaks, we observed some secondary fine structures in some spectra measured with magnetic field. For example, these fine structures can be clearly visualized in Fig.5c. We can probably attribute these fine structures to the 2D electron states. But clearly they are much weaker comparing with the intensity of the main structure arising from the LL peaks of the surface Dirac electrons. Similar fine structures were also observed in the Landau level studies of Bi₂Se₃ by STM (Phys. Rev. B 82, 081305(R) (2010)) and transport measurements (PRL 103, 246601 (2009)). The authors there approved that the pronounced LL peaks are originated from the topologically protected surface state and the fine structures with smaller amplitude may originate from another 2D electron state. Therefore, we have revised the paper in related places by following this suggestion. Since the LL peak effect is a mixing consequence of the topological surface states and the 2D EG, although the former is the dominant one, we still agree with the referee to change the wording "smoking gun" to "is consistent with TSS". We have added the discussion to the text (see second half on page 11).

2) A possible typo in line 153: is it really Fig. 2c?

Response: We thank the referee for pointing out this error. It should be Fig.2d. This has been corrected.

Reviewer #2 (Remarks to the Author):

The authors have substantially edited their manuscript and taken on board many of the suggestions. The manuscript has improved significantly, and there are only a few points which I think should still be accounted for. I recommend publication in Nature Communications after the suggested changes have been considered.

The discussion of percentage of density of states of gaps for isotropic s-wave and anisotropic s-wave is still confusing, I believe this is mainly due to the notation the authors use. I recommend introducing different symbols for the gap functions themselves (which are currently called Δ_s and $\Delta_{s\text{-ani}}$) and the associated densities of states.

Response: We thank the referee for this comment. The notations about the fitting gaps and the associated density of states have been modified and clearly expressed in the revised version. Our fitting is actually based on a two-component model (s wave and ani-s wave). We have changed the notions and description throughout the paper. We regret for not make it more clearly in the original version.

A few points need clarification:

* The Landau levels themselves are not the smoking gun that the surface state becomes gapped, but much rather the fact that the gap goes to zero rather than to a finite value.

Response: Indeed, the Landau levels themselves are not the smoking gun evidence that the surface state becomes gapped. We regret for not having made this point better expressed in the old version. We observed a much weakened amplitude of the LL oscillations within the gap. This can be understood as the consequence of the pairing and condensing of the surface Dirac electrons.

* For the averaging of the spectra in fig. 5, two different energy ranges are mentioned: the main text says $[-1,1]$, whereas the caption indicates $[-0.5,0.5]$. This average should also be plotted in fig. 5a, together with error bars.

Response: We thank the referee for pointing out this error. The correct range for averaging the spectra should be $(-1\text{mV}, 1\text{mV})$. This was mistakenly written as $(-0.5\text{mV}, 0.5\text{mV})$ in the caption of Fig.5. Now this has been corrected.

* In the caption of figure 4, "MV" should be defined.

Response: We thank the referee for this suggestion. 'MV' is the abbreviation for 'mean value' as indicated in the main text. We have replaced 'MV' by 'mean value' in the caption

of figure 4 for clarification.

* The additional figures in the supplementary material showing the various fits are very good. Fig. S4b-d make one wonder whether the fit with a slightly larger value of Gamma is maybe better than the one shown in fig. 2d ?

Response: We thank the referee for this suggestion. Fig.S4 shows the comparison of two component fitting (s wave and anti-s wave) and single s wave fitting using different Γ values. It turns out that the single s wave fitting can't track the low energy feature and the coherence peaks on the STS at the same time. The fitting in Fig.S4d is indeed more close to the raw data comparing with the old fitting curve shown in Fig.2d, although the low energy part still has a deviation. We have replaced the fitting curve in Fig.2d by the one in Supplementary Fig.4d. Some modifications have also been made in related places, including the values in Supplementary Table 1.

The comments of the second referee:

The revised version of the manuscript has accounted for almost all changes both referees had suggested.

One point which the authors probably overlooked but which I believe is important is that in fig. 5a a plot of the mean value of the averaged Landau level oscillations in the window $[-1\text{mV}, 1\text{mV}]$ (with error bars) is still missing (next to the one for the window $[-10\text{mV}, 10\text{mV}]$). To be able to follow the authors on how they obtain the ratio in fig. 5b this would be important. Given that this ratio is close to one, the mean value of these oscillations should fit rather nicely in fig. 5a.

With this point fixed, I recommend publication in Nature Communications.

Responses:

We appreciate the careful review and positive response of this referee. Indeed, in last version, we overlooked and thus forgot to plot the “averaged Landau level oscillations in the window $[-1\text{mV}, 1\text{mV}]$ (with error bars)”. Here we plot them together with the data in the window $[-10\text{mV}, 10\text{mV}]$. The ratio between them keeps the same and is shown as before in Fig.5b. Here we adopt a standard method based on Gaussian distribution analysis to determine the error bars, which is detailed in the Supplementary Information.